# Brain Connectivity Affecting Gait Function after Unilateral Supratentorial Stroke

**DOI:** 10.3390/brainsci11070870

**Published:** 2021-06-29

**Authors:** Hyun-Ah Lee, Dae-Hyun Kim

**Affiliations:** 1Department of Physical Medicine and Rehabilitation, Veterans Health Service Medical Center, Seoul 05368, Korea; hyuna5575@gmail.com; 2Department and Research Institute of Rehabilitation Medicine, Yonsei University College of Medicine, Seoul 03722, Korea

**Keywords:** stroke, gait, connectivity, neurorehabilitation, functional recovery

## Abstract

Gait dysfunction is a leading cause of long-term disability after stroke. The mechanisms underlying recovery of gait function are unknown. We retrospectively evaluated the association between structural connectivity and gait function in 127 patients with unilateral supratentorial stroke (>1 month after stroke). All patients underwent T1-weighted, diffusion tensor imaging and functional ambulation categorization. Voxel-wise linear regression analyses of the images were conducted using fractional anisotropy, mean diffusivity, and mode of anisotropy mapping as dependent variables, while the functional ambulation category was used as an independent variable with age and days after stroke as covariates. The functional ambulation category was positively associated with increased fractional anisotropy in the lesioned cortico-ponto-cerebellar system, corona radiata of the non-lesioned corticospinal tract pathway, bilateral medial lemniscus in the brainstem, and the corpus callosum. The functional ambulation category was also positively associated with increased mode of anisotropy in the lesioned posterior corpus callosum. In conclusion, structural connectivity associated with motor coordination and feedback affects gait function after stroke. Diffusion tensor imaging for evaluating structural connectivity can help to predict gait recovery and target rehabilitation goals after stroke.

## 1. Introduction

Gait dysfunction is one of the most common symptoms after stroke, affecting 26–33% of stroke victims. It may present during the first 1–15 years after stroke [1,2]. Recovery of gait function can decrease dependency on caregivers in performing activities of daily living. Hence, a better understanding of the brain areas affecting gait function can help physicians to plan an effective acute neurorehabilitation program after stroke.

The relationship between neuronal connectivity and gait function after stroke remains unclear. Gait function is one of most complex motor skills in humans, and the control of gait relies on multiple brain areas, including the cerebral cortex, brainstem, cerebellum, and basal ganglia [3]. Previous studies using neuroimaging have attempted to describe the relationship between the location of the stroke lesion and gait dysfunction [4,5,6,7]. Lesions in the sensorimotor, premotor cortex, supplementary motor area in the cerebral cortex, and cerebellum have been correlated with poor gait function after stroke [7,8,9]. These studies only assessed the direct effects of lesions primarily in cortical areas, and control of human gait depends on the connections and coordination of multiple brain regions. For example, Jang et al. showed a positive correlation between the integrity of the corticoreticular pathway in white matter and gait function after stroke [4]. However, this study did not consider all white matter connectivity but only evaluated the specific role of the corticoreticular pathway. Since the execution of human gait involves multiple cortical brain areas as well as the cerebellum, disruption of their connections by stroke likely affects gait function. Hence, previous studies could not elucidate a definitive association between brain connectivity and gait function after stroke.

Diffusion tensor imaging (DTI) is a widely used method for investigating the connectivity of white matter tracts [10]. The values calculated using DTI can reflect various aspects relating to white matter conditions. Fractional anisotropy (FA) indicates parallel movement of water molecules in fiber tracts, giving white matter cylindrical anisotropic properties. Mean diffusivity (MD) reflects the average water movement without any directionality, and increased MD indicates vasogenic edema and axonal loss. The mode of anisotropy (MO) value indicates a difference in the shape of anisotropy ranging from planar (crossing fibers) to linear (one fiber predominant) [10]. Here, we used FA, MD, and MO values from DTI to investigate white matter connectivity in brain structures associated with gait function after supratentorial stroke.

## 2. Materials and Methods

### 2.1. Patients and Study Design

This study retrospectively reviewed the medical records of 384 patients who underwent post-stroke rehabilitation and DTI between May 2015 and May 2020. The inclusion criteria were (a) first ever stroke, as confirmed by computed tomography or magnetic resonance imaging (MRI); (b) supratentorial stroke; (c) >1 month after stroke onset; (d) Mini-Mental Status Examination score > 20; (e) MRI and functional ambulation category (FAC) evaluation performed within a one-week interval after stroke; (f) gait independently assessed on stairs and inclines before stroke. The exclusion criteria were (a) quadriplegia or double hemiplegia; (b) any coexisting orthopedic or neurological disease that could affect gait function. One hundred twenty-seven post-stroke patients met the criteria for this study. The study protocol was approved by our Institutional Research Ethics Committee for Human Subjects, which waived the requirement for informed consent due to the retrospective nature of the study.

### 2.2. Mini-Mental Status Examination

The Mini-Mental Status Examination was designed as a simplified scored form of the cognitive mental status examination which includes eleven questions, and is therefore practical to use serially and routinely [11].

### 2.3. Functional Ambulation Category

The FAC is a functional walking test that evaluates gait ability. It consists of a six-point scale to assess ambulation status by determining how much caregiver support the patient requires when walking, regardless of whether or not they use a personal assistive device (FAC 5: ambulates independently on stairs and inclines; 4: ambulates independently on a level surface; 3: ambulates with standby of one person; 2: requires intermittent support of one person; 1: requires continuous support of one person; 0: cannot ambulate or requires the help of more than one person) [12].

### 2.4. MRI Acquisition and Preprocessing

All MRI scans were acquired using a 3-T MR scanner (Siemens, Erlangen, Germany) with a 20-channel head coil. High-resolution 3D T1-weighted images and DTI scans were obtained. T1-weighted parameters were: TR (repetition time)/TE (echo time) = 1900/2.57 ms, matrix = 256 × 256, field of view = 230 × 230 mm^2^, flip angle = 9°, and slice thickness = 1 mm. DTI parameters were: TR/TE = 9700/92.0 ms, matrix = 112 × 112, field of view = 224 × 224 mm^2^, number of excitations = 1, 30 directions, b = 1000 s/mm^2^, and slice thickness = 2 mm. For measurement of lesion volume and voxel-based lesion symptom mapping, a neuroradiologist drew the lesion of each patient on T1-weighted images in the native space using 3D Slicer software (version 4.8.1; https://www.slicer.org, accessed on 5 May 2018). For further analysis, MRIs from 61 patients with right hemispheric lesions were flipped horizontally to identify the corresponding non-lesioned area in the left hemisphere. After this inversion, diffusion-weighted images were registered to the corresponding b = 0 image with an affine transformation to correct distortions caused by eddy currents. The FA, MO, and MD maps were constructed in a native space. Subsequently, T1-weighted images, FA, MO, and MD maps were nonlinearly transformed to the standard Montreal Neurological Institute space.

### 2.5. Statistical Analysis

For entire brain areas, voxel-wise linear regression analysis was performed using transformed FA, MO, and MD maps as dependent variables, FAC as the independent variable, and age and duration of dysfunction after stroke as covariates. To identify brain areas with significant involvement, statistical analyses were performed at a cluster-level, family-wise error (FWE)-corrected threshold of *P*_FWE_ < 0.05 with a primary cluster-defining threshold of uncorrected *P* < 0.001 [13].

## 3. Results

### 3.1. Patients’ Characteristics

The mean age of the patients was 73.2 ± 7.1 years (range 48.5–95.5) and the duration of gait disturbance after stroke was 1076.2 ± 1782.2 days (range 32.0–9521.0). One hundred five patients had an ischemic stroke, and twenty-two patients had hemorrhagic stroke. Sixty-one lesions were in the right hemisphere and sixty-six were in the left hemisphere. Table 1 summarizes the general and clinical characteristics of all patients and Figure 1 summarizes the lesion data.

### 3.2. Brain Areas Associated with Functional Ambulation Category

The FAC was positively associated with increased FA values of the cortico-ponto-cerebellar system (CPCS), such as the corticopontine fiber pathway at the lesioned midbrain level and non-lesioned middle cerebellar peduncle (*P*_FWE_ < 0.05). The FAC was also positively associated with increased FA values of the corona radiata, which passes through the corticospinal tract (CST) in the non-lesioned hemisphere; the medial lemniscus at the bilateral midbrain, which passes through the main sensory tract, and the corpus callosum (*P*_FWE_ < 0.05). The FAC was positively associated with increased MO values of the lesioned posterior corpus callosum (*P*_FWE_ < 0.05). There was no significant brain area negatively associated with FA and MO values. Furthermore, no significant brain area showed positive or negative association with MD values (Figure 2 and Table 2).

## 4. Discussion

In this study, we found that the connectivity of multiple brain regions was associated with gait function after stroke. Higher FA value from DTI indicates increased parallel movement of water molecules to fiber tracts, which corresponds to increased structural connectivity [14,15]. Increased structural connectivity of the CPCS and CST is positively associated with gait function. Higher MO values show that one fiber predominates rather than indicating the presence of crossing fibers [10,16]. Single-fiber predominance in the lesioned posterior corpus callosum was also positively associated with gait function. Higher MD values indicate increased axonal and myelin loss [14]. However, there were no significant brain regions affected, relative to axonal loss and gait function in bilateral hemispheres. This finding has potential implications for understanding the mechanism of gait recovery after stroke, the utility of DTI for prediction of gait function, and the identification of the most productive rehabilitation goal.

The CPCS is a major neural circuit of the cerebellum involved in movement coordination [17]. The corticopontine tract, originating from the cortical brain regions, projects to the contralateral cerebellar cortex via descending fibers which cross in the pons and pass through the contralateral middle cerebellar peduncle [18]. The connectivity along the corticopontine tract critically influences cerebellar activity [18,19]. Its disruption can suppress cerebellar neuronal activity [7,18], which is usually accompanied by cerebellar symptoms including ataxia [17]. Hence, increased structural connectivity in the pathway of lesioned corticopontine tracts at the midbrain level and the middle cerebellar peduncle may affect gait function by reducing ataxia and enhancing coordination during gait.

The role of the CST of the non-lesioned hemisphere in lower limb function is still unknown. Activation of the CST in the non-lesioned hemisphere has been associated with worse outcomes in upper limb function [20,21]. Unlike for upper limb function, reorganization of neural connections underlying lower limb function have not been characterized after stroke. In comparing the use patterns of the upper (primarily unilateral reaching and hand motion) and lower (mainly bilateral walking) limbs and the corresponding differences in brain activation associated with these movements [22,23], a transfer of the findings from the upper limb to lower limb seems unjustified. Bilateral innervation of axial and lower limb muscles also affects different roles of the CST of the non-lesioned hemisphere in recovering gait function [21]. The intact CST may compensate for the injured CST in rescuing lower limb function and gait after stroke, because of both bilateral innervation and reserve of reorganization. The increase of connectivity in intact CST, especially at the level of the corona radiata, may enhance gait function after stroke.

The medial lemniscus system consists of mechanoreceptors in the skin, muscle spindles, tendon organs, and other proprioceptive stimuli. Bilateral proprioceptive input from the lower limbs affects gait function as it travels through the medial lemniscus to the midbrain. Hence, injury to the medial lemniscus causes impairment of contralateral proprioception in the upper and lower limbs [24]. This results in the loss of sensory feedback to the brain concerning the current positions of the lower limbs, causing gait instability. Medial lemniscus disruption also causes a distinctive form of ataxia. Retention of postural equilibrium during gait can be accomplished through proprioceptive information processed by the vestibular system and cerebellum [25].

The corpus callosum connects the right and left cerebral hemispheres of the brain and is an important neural structure controlling and coordinating bilateral movements. Additionally, the corpus callosum’s major connections are organized by anatomical portions. The most anterior portion of the callosum (genu) connects the prefrontal cortices on either hemisphere. The middle portions of the corpus callosum connect motor and somatosensory regions. The caudal part of the body of the corpus callosum and the most posterior section of the corpus callosum (splenium) connect the cortex from the temporoparietal-occipital junction. The splenium also connects the dorsal parietal and occipital regions [26,27]. Gait incorporates a specific set of movements that involve coordination of the lower limbs [25]. The corpus callosum is essential for the effective control of these complex movements [25]. For the precise temporal and spatial coordination between 2 sides of body, movement of one limb has an transcallosal inhibition on the ipsilateral motor cortex [28,29]. And the amount of transcallosal inhibition increases as bilateral movements become more complex and less synchronous [28]. This transcallosal inhibition reduces the interference which can arise due to motor overflow or the interhemispheric excitation which cause unintended activation in the contralateral motor cortex [25]. So, the increase of connectivity in corpus callosum may enhance gait coordination after stroke. And decreased connectivity in the corpus callosum was shown to influence transcallosal communication between bilateral cortical hemispheres, thereby disrupting the transfer and integration of motor and sensory information [30,31]. The results of this study are in line with recent studies of significant associations between callosal structural integrity and gait dysfunction [32,33]. Because gait function is influenced by broad cortical areas including premotor, primary motor, primary sensory, sensory association and visual cortices, the connectivity of corpus callosum throughout the anterior to posterior portion may affect gait function.

A few limitations should be considered when interpreting the results. First, this study had a cross-sectional and retrospective design. Second, the sample was heterogenous; most subjects were men because of our institutional characteristics. Third, the duration after stroke was heterogeneous; even though the subjects’ onsets of stroke were more than 1 month, durations after stroke varied from a month to several years. Additionally, patients who were recovering from a neurologic injury could be included in this study. Therefore, heterogenicity of onset of strokes could affect the outcome. Furthermore, depending on the type of stroke (ischemic or hemorrhagic), the recovery process may vary. Fourth, the heterogenicity of stroke lesion volume could also affect the outcome. Despite these limitations, this is the first study to analyze the association between structural connectivity and gait function after stroke.

## 5. Conclusions

The structural connectivity of brain regions underlying motor control and coordination, as well as sensory feedback, has a critical role for maintaining gait after stroke. DTI may be useful to evaluate structural connectivity and predict gait function changes in individual patients after stroke. Furthermore, the evaluation of changes in structural connectivity can help determine the appropriate target for neuromodulation for recovery of gait function and help formulate a personalized rehabilitation goal.

## Figures and Tables

**Figure 1 brainsci-11-00870-f001:**
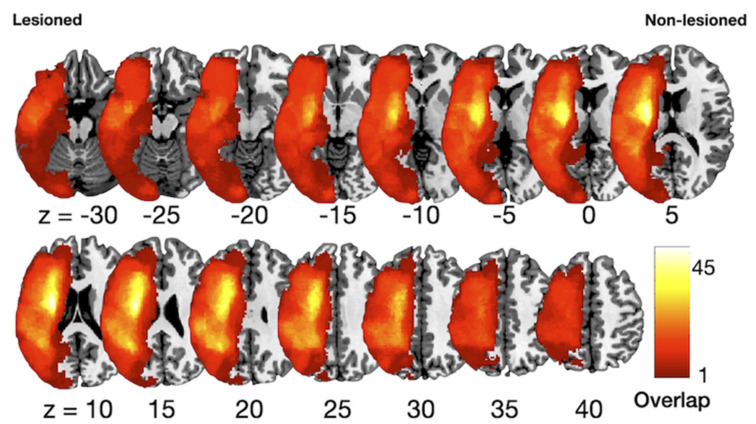
Lesion overlapping maps. A T1-weighted template was used to demarcate lesions for every patient. The color scale indicates the percentage of overlapping lesions across patients.

**Figure 2 brainsci-11-00870-f002:**
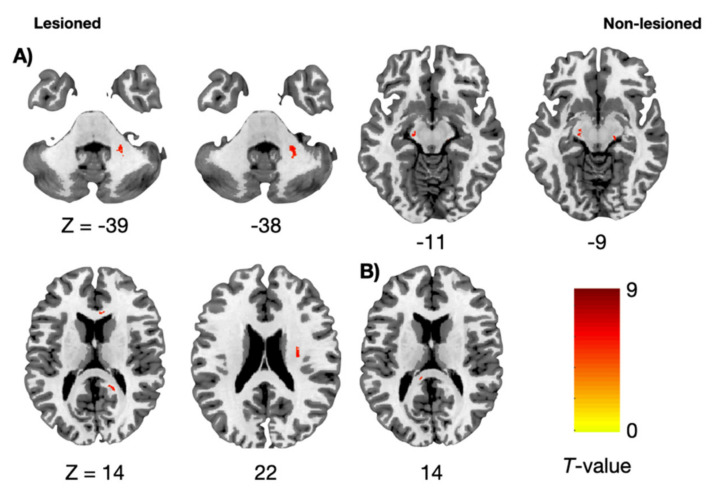
Regression analysis associated with functional ambulation category. (**A**) Fractional anisotropy maps associated with functional ambulation category. (**B**) Mode of anisotropy maps associated with functional ambulation category. z: *z*-axis in the Montreal Neurological Institute space. Statistical threshold: *P*_FWE_ < 0.05.

**Table 1 brainsci-11-00870-t001:** General characteristics and functional ambulation category of included patients.

Variables	Patient
Demographic characteristics	
Age (years, mean ± SD)	73.2 ± 7.1
Sex (male:female)	124:3
Duration after stroke (day, mean ± SD)	1076.2 ± 1782.2
Mini-Mental Status Examination (mean ± SD)	25.8 ± 3.4
Stroke type (number, %)	
Anterior cerebral artery infarction	7 (5.5%)
Middle cerebral artery infarction	87 (68.5%)
Posterior cerebral artery infarction	5 (3.9%)
Watershed infarction (between ACA and MCA)	2 (1.6%)
Lacunar infarction	4 (3.2%)
Basal ganglia hemorrhage	9 (7.1%)
Thalamic hemorrhage	9 (7.1%)
Frontoparietal hemorrhage	4 (3.2%)
Lesioned hemisphere (Rt:Lt)	61:66
Lesion volume (cm^3^)	471.6 ± 823.6
Functional ambulation category (number, %)	
0	20 (15.8%)
1	13 (10.2%)
2	19 (15.0%)
3	25 (19.7%)
4	34 (26.8%)
5	16 (12.6%)

Key: SD: standard deviation; Rt: right; Lt: left.

**Table 2 brainsci-11-00870-t002:** Result of regression analysis associated with functional ambulation category.

Regions	Side	Peak MNI Coordinate		Cluster	Max T		Value
		x	y	z	Size		
middle cerebellar peduncle	Non-lesioned	23	−43	−38	73	5.97	FA
midbrain	Lesioned	−17	−15	−11	57	6.27	FA
midbrain	Non-lesioned	17	−21	−9	59	6.27	FA
anterior corpus callosum	Non-lesioned	11	29	0	57	6.08	FA
anterior corpus callosum	Non-lesioned	12	32	9	62	5.71	FA
posterior corpus callosum	Lesioned	−20	−52	12	51	6.07	FA
corona radiata	Non-lesioned	15	−45	14	50	5.84	FA
corona radiata	Non-lesioned	30	−13	22	81	6.02	FA
corona radiata	Non-lesioned	13	11	28	32	5.66	FA
corpus callosum	Lesioned	−10	−36	14	23	5.80	MO

The cluster size refers to the voxel number. The Montreal Neurological Institute (MNI) coordinates of the peak of clusters (x, y, z) are represented. Max T indicates the maximal *t* value in the multiple regression test. Key: FA: fractional anisotropy; MO: mode of anisotropy.

## Data Availability

The data presented in this study are available on request from the corresponding author. The data are not publicly available due to privacy.

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
