# Peer review of "Brain Connectivity Affecting Gait Function after Unilateral Supratentorial Stroke"

_brainsci, 2021, doi:10.3390/brainsci11070870_

Round 1

Reviewer 1 Report

This interesting study by Hyun Ah Lee and Dae Hyun Kim assesses the association between structural connectivity (analyzed with DTI metrics) and gait function in patients with a unilateral stroke.

The study is well written.

In paragraph 2.1, could you explain what "Mini Mental Examination Score" refers to? And please cite Folstein et el., 1975, J. Psychiatr. Res..

In paragraph 2.3, you state that there were 62 patients with a lesion on the right hemisphere but in table 1 the number is 61. Could you please check?

Is there any published data showing an association between neuromodulation and structural connectivity? If so, could you discuss?

Author Response

This interesting study by Hyun Ah Lee and Dae Hyun Kim assesses the association between structural connectivity (analyzed with DTI metrics) and gait function in patients with a unilateral stroke.

The study is well written.

Thank you for giving us an opportunity to revise our manuscript. We hope our revision of the manuscript will be acceptable.

Point 1: In paragraph 2.1, could you explain what "Mini Mental Examination Score" refers to? And please cite Folstein et el., 1975, J. Psychiatr. Res..

Response 1: We have added the paragraph 2.2 Mini-Mental Status Examination. As the reviewer suggested, we have added the citation.

Point 2: In paragraph 2.3, you state that there were 62 patients with a lesion on the right hemisphere but in table 1 the number is 61. Could you please check?

Response 2: We thank the reviewer for this comment. We have checked the number, and total number of patients with a lesion on the right hemisphere is 61 patients. We corrected the number.

Point 3: Is there any published data showing an association between neuromodulation and structural connectivity? If so, could you discuss?

Response 3: We thank the reviewer for this comment. Neuromodulations for dysphagia are rTMS (repetitive transcranial magnetic stimulation) and tDCS(Transcranial direct current stimulation). We have searched articles showing association between neuromodulation (rTMS and tDCS) and structural connectivity, but we only found one case study. We think it is difficult to revise the discussion with one case study.

Reviewer 2 Report

In their manuscript ‘Brain connectivity affecting gait function after unilateral supratentorial stroke’, authors Lee and Kim have investigated recovery of gait function in stroke patients using retrospective analyses of brain images. They find that fiber tract connectivity in the brain is associated with motor coordination and recovery of gait after stroke is affected by feedback mechanisms through the connecting structures.

The study is well designed and well described. The authors’ conclusions are supported by their analyses. The authors have appropriately used tables and figures to explain their results. This study could be useful in designing targeted rehabilitation regimen for individuals after stroke.

Author Response

In their manuscript ‘Brain connectivity affecting gait function after unilateral supratentorial stroke’, authors Lee and Kim have investigated recovery of gait function in stroke patients using retrospective analyses of brain images. They find that fiber tract connectivity in the brain is associated with motor coordination and recovery of gait after stroke is affected by feedback mechanisms through the connecting structures.

The study is well designed and well described. The authors’ conclusions are supported by their analyses. The authors have appropriately used tables and figures to explain their results. This study could be useful in designing targeted rehabilitation regimen for individuals after stroke.

Response: We appreciate for your comments. As you mentioned, our study can be useful in designing targeted rehabilitation regimen for individuals after stroke.